# From Eww to Woo: Detection of Mental Health Disturbing Images in Social Media

## Abstract

Exposure to distressing images on social media, such as gore and other graphic content, can lead to significant mental health issues and disturbances. This paper introduces a novel dataset specifically curated to include such harmful images, aiming to facilitate the development of machine learning models capable of detecting and filtering these types of content. By training on this dataset, the proposed models demonstrate the ability to accurately identify and flag disturbing images, thereby contributing to the mitigation of mental health risks associated with prolonged exposure to harmful visual content on social media platforms. The proposed dataset is benchmarked on various state of the art models with the accuracy 70.15%. This work represents a critical step towards creating safer online environments and protecting users' mental well-being.

**The paper contains potentially mental health disturbing images.**

## 1 Introduction

Social media platforms provide an excellent means of connecting with people across the globeBoyd & Ellison (2007), fostering communication and community in ways previously unimaginableFardouly et al. (2015). These platforms enable users to share experiences, stay informed, and engage with diverse perspectives, making the world feel more interconnected than ever before. However, alongside these benefits lies the potential for exposure to harmful content. One particularly concerning issue is the continuous viewing of certain images, which can negatively impact mental healthGillespie (2018); Ghosh & Anwar (2021). The constant barrage of idealized images and distressing visuals can lead to feelings of inadequacy, anxiety, and depression, highlighting the darker side of our digital interactionsGorwa et al. (2020).

Some social media platforms have implemented restrictions, such as NSFW (Not Safe For Work) filters, to limit exposure to explicit or inappropriate contentHampton & Wellman (2003); Qayyum et al. (2024). These measures aim to create a safer online environment and protect users from potentially harmful material. By flagging or hiding content deemed inappropriate, platforms strive to reduce the likelihood of users encountering distressing images or videos. These restrictions reflect a growing recognition of the need for responsible content management in an era where digital media consumption is ubiquitousLewis & Seko (2016); Liu et al. (2021a). Despite these efforts, users can still encounter images that may disturb their mental health. The continuous exposure to idealized lifestyles, graphic content, or distressing visuals can have a profound impact on individuals. For instance, constantly viewing images that portray unrealistic body standards or extravagant lifestyles can lead to feelings of inadequacy, jealousy, and low self-esteem. Similarly, exposure to violent or traumatic content can induce anxiety, stress, or even trigger past traumas. This highlights the ongoing challenge of ensuring a truly safe and supportive digital space, as the effectiveness of content restrictions is often limited by the sheer volume and variety of content shared online. Additionally, the algorithms used to detect harmful content are not foolproof and may fail to identify all problematic material, further complicating efforts to protect users' mental healthPrimack et al. (2017).

**Motivation:** The rapid expansion of social media has revolutionized connectivity and information sharing but has also exposed users to distressing and harmful visual content, such as gore and graphic scenes, which pose significant mental health risks including anxiety, depression, and PTSD. Current detection systems are inadequate due to the complexity and variability of such content, compounded

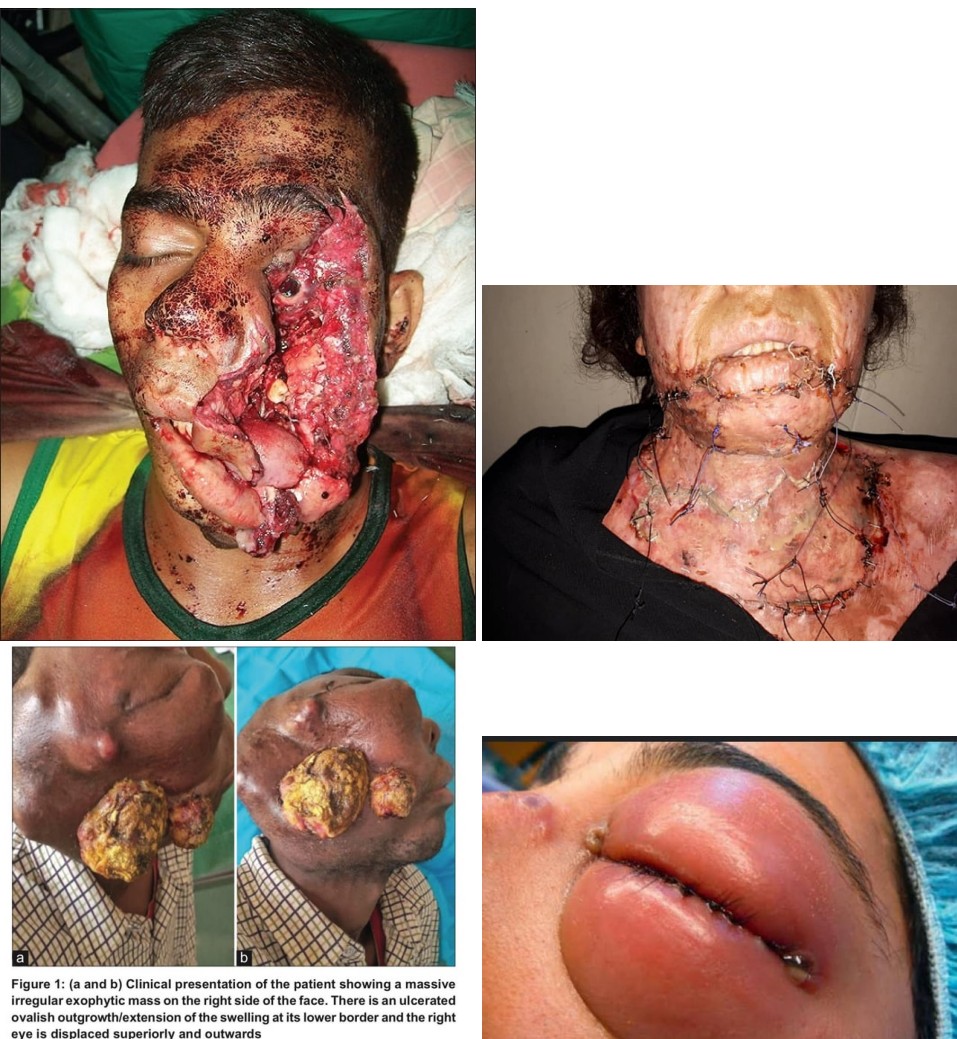

Figure 1: (a and b) Clinical presentation of the patient showing a massive irregular exophytic mass on the right side of the face. There is an ulcerated ovalish outgrowth/extension of the swelling at its lower border and the right eye is displaced superiorly and outwards

Figure 1: Examples of the images in social media those disturb mental health

by the lack of high-quality, curated datasets essential for training robust models. This paper introduces a novel dataset specifically curated to include harmful images, addressing this critical gap and facilitating the development of advanced machine learning models capable of accurately identifying and filtering distressing content. By enhancing the detection of harmful images, this work aims to create safer online environments and protect users' mental well-being.

The main contributions of the paper are:

1. We propose a novel dataset for the detection of the images that are mentally disturbing in the social media.

2. We have implemented the state of the art models and presented the results.

## 2 RELATED WORKS

In the realm of content detection on social media platforms, numerous studies have addressed the challenge of identifying and filtering inappropriate content. Yousaf & Nawaz (2022) proposed a deep learning-based architecture for the detection and classification of inappropriate content in YouTube videos. Their framework employed an ImageNet pre-trained convolutional neural network (CNN) model, EfficientNet-B7, to extract video descriptors, which were then fed into a bidirectional long

short-term memory (BiLSTM) network to learn effective video representations. An attention mechanism was integrated to enhance the model's focus on critical features. Evaluated on a manually annotated dataset of 111,156 cartoon clips, their EfficientNet-BiLSTM model achieved an impressive accuracy of 95.66%, demonstrating the superiority of deep learning classifiers over traditional machine learning techniques. This work highlights the efficacy of CNN and LSTM architectures in handling video content classification tasks, setting a benchmark for future research in this domain.

Mazumdar et al. (2024) proposed a novel dataset to detect mental health disorders during the metaverse and the youth addicting to the virtual reality. Papadamou et al. (2019) conducted a large-scale study on toddler-oriented disturbing content on YouTube, developing a classifier with 82.8% accuracy. They found that inappropriate videos are prevalent on the platform and current countermeasures by YouTube are inadequate. This highlights the necessity for advanced detection mechanisms to protect young viewers from harmful content. Papadamou et al. (2020) investigated the Elsagate phenomenon, proposing a deep learning model trained on a dataset of 3K videos to detect such content. This study emphasizes the need for more refined approaches to mitigate exposure to harmful content on YouTube.

Recent studies have explored the utilization of social media data for event detection due to the vast amount of user-generated content and real-time nature of these platforms. It provide a comprehensive review of trends in event detection by analyzing social media data, systematically examining 67 articles from the past decade. They categorize existing event detection techniques, including shallow machine learning, deep learning, and rule-based approaches, and highlight the primary challenges such as language compatibility, data heterogeneity, and real-time processing. The review also underscores the necessity for robust models that can handle diverse media types and dialects, aiming to enhance the accuracy and efficiency of event detection systems Mredula et al. (2022).

Another study developed and validated an automatic image-recognition algorithm aimed at detecting intentional self-harm, specifically cutting-related posts, on Instagram. Utilizing convolutional neural networks, the algorithm achieved an 87%accuracy rate in distinguishing between NSSI and non-NSSI content. The study involved scraping Instagram posts tagged with #cutting, #suicide, and their German equivalents over a 48-hour period, analyzing 13,132 images. The results highlight the algorithm's potential for real-time monitoring and intervention, providing a significant step forward in using machine learning for mental health research and prevention on social media platforms Scherr et al. (2019).

A recent study developed a novel approach for detecting fire in social media images through the Fast-Fire Detection (FFireDt) method. This approach leverages instance-based learning, combining feature extraction methods and evaluation functions to achieve high precision in fire detection. The study introduced the Flickr-Fire dataset, an annotated set of images depicting fire occurrences, and evaluated 36 different image descriptors. The results demonstrated that FFireDt could achieve a precision comparable to human annotators, providing a significant advancement in automatic image analysis for crisis management. This work lays the foundation for future developments in monitoring and analyzing social media images for effective crisis response Bedo et al. (2015).

Marra et al. (2018) addressed the challenge of detecting GAN-generated fake images on social networks, emphasizing the increasing threat posed by such realistic fakes. The study evaluates several image forgery detectors against image-to-image translation manipulations, including conventional techniques and deep learning approaches. Their findings reveal that while conventional detectors can achieve detection accuracies up to 95% in ideal conditions, deep learning methods maintain high accuracy, up to 89%, even on compressed images typically found on social networks. This research highlights the critical role of robust detection algorithms in mitigating the impact of sophisticated fake images in digital media.

Won et al. (2017) introduce a novel visual model for detecting protest activities and estimating perceived violence from social media images. The model employs a multi-task convolutional neural network to classify the presence of protesters, predict visual attributes, and estimate perceived violence and emotions in images. Their study spans major protest events from 2013-2017, utilizing a dataset of 40,764 images. The model demonstrates effective performance in classifying protest activities and estimating perceived violence, providing a new tool for analyzing protest dynamics from visual data. This research contributes significantly to understanding social media's role in character-

izing real-world protests and offers insights into the visual sentiments and attributes associated with these events.

This work stands out due to its introduction of a novel dataset specifically curated to include a diverse range of mentally disturbing images, such as gore and graphic content, which comprehensively represent the types of disturbing images users encounter on social media. Unlike existing datasets, which often lack the breadth and specificity needed for effective harmful content detection. The benchmarking of our dataset on various state-of-the-art models showcases its utility and effectiveness, with our proposed models demonstrating a significant ability to flag mentally disturbing images accurately.

## 3 METHODOLOGY

Data was collected from Reddit and Google utilizing the Selenium Chrome web driver. Specifically, data was extracted from the Reddit channel r/medicalgore and from various Google search results. The images were collected over the period from January 2021 to June 2024.

**Data Annotation:**The primary objective of the annotation process was to classify images as either mentally disturbing or non-disturbing. Three annotators, each holding a Master's degree in psychology with first class grades in all subjects, carried out the annotation. These annotators also have relevant work experience in mental health and social media.

To minimize labeling errors, the dataset was divided into batches and each data point was labeled by at least two annotators. If both annotators agreed on a label, it was accepted as final. In cases where there was disagreement, the annotators discussed the image until a consensus was reached.

### 3.1 ANNOTATION GUIDELINES

The primary objective of the annotation process is to classify images as either mentally disturbing or non-disturbing. To achieve this, we employed three annotators, each holding a Master's degree in psychology and possessing relevant work experience in mental health and social media. The following guidelines were established to ensure consistency and reliability in the annotation process:

**Classification Criteria**:

- **Mentally Disturbing**: Images likely to cause psychological discomfort, anxiety, or distress.
- **Non-Disturbing**: Images that do not evoke significant negative psychological responses.

**Annotation Process**: Each image is reviewed independently by two annotators who assign labels based on their initial assessment. If the annotators' labels differ, a discussion is initiated to resolve discrepancies. In cases where consensus cannot be reached after discussion, a third annotator is consulted.

**Labeling Considerations**:

- **Context**: Consider the context in which the image appears, as an image that might be disturbing in one context may not be in another.
- **Content**: Focus on the visual content, such as the presence of blood, injuries, or other graphic elements.
- **Emotional Impact**: Assess the potential emotional impact on an average viewer without specialized training in medicine or psychology.

**Quality Control**: Regular cross-checks and discussions among annotators are conducted to ensure consistency in labeling. Periodic reviews of annotated batches are performed to identify and rectify any discrepancies or errors.

**Ethical Considerations**: Annotators should be aware of their own psychological state and take breaks as needed to prevent desensitization or emotional distress. Additionally, images should be anonymized to protect the privacy of individuals depicted.

Table 1: Dataset Overview and Data Statistics

(a) Overview of the dataset

| Image Path | Label |
|------------|-------|
| a_1189.jpg | 1 |
| a_891.jpg | 0 |
| a_284.jpg | 1 |
| a_1018.jpg | 0 |

(b) Data Statistics. 1 represents mental health disturbing images, 0 represents images that do not

| Metric | 0 | 1 | Overall |
|--------|------|------|---------|
| Train set | 6702 | 6630 | 13332 |
| Test set | 2234 | 2210 | 4444 |
| Total | 8936 | 8840 | 17776 |

By adhering to these guidelines, we aim to ensure a reliable and consistent annotation process, thereby contributing to the robustness and validity of our research findings.

## 3.2 ANALYSIS OF THE DATASET

Table 1(a) provides an overview of the dataset with sample entries, illustrating the image paths and their corresponding labels. This foundational information helps to understand how the data is organized and sets the stage for a deeper analysis. The detailed breakdown of the dataset is presented in Table 1(b), which includes the number of images in each category (0 or 1) for both the training and test sets. This breakdown reveals several key insights. Firstly, the overall dataset consists of 17,776 images, with 8,936 images labeled as non-disturbing and 8,840 images labeled as mentally health disturbing. This indicates a nearly balanced dataset in terms of the overall distribution of labels, which is essential for creating unbiased machine learning models.

Focusing on the training set, we find that it contains 13,332 images, with 6,702 images labeled as non-disturbing and 6,630 images labeled as disturbing. This training set represents approximately 75% of the total dataset, maintaining a near-equal distribution between the two categories. Such balance is beneficial as it ensures that the model receives a well-rounded exposure to both types of images during the learning phase. The test set, comprising 4,444 images, includes 2,234 images labeled as non-disturbing and 2,210 images labeled as disturbing. This test set accounts for about 25% of the total dataset and also maintains a balanced distribution of labels. The consistency in the distribution between the training and test sets is crucial for evaluating the model's performance reliably. It helps to ensure that the model's accuracy and generalization capabilities are assessed without any inherent bias towards one category.

## 3.3 BASELINE IMPLEMENTATION

The baseline implementation involves binary classification of images using the proposed dataset. The following models were implemented: (i) ViTDosovitskiy et al. (2021), (ii) DeiTTouvron et al. (2021), (iii) Swin TransformerLiu et al. (2021b), (iv) TNTHan et al. (2021), (v) CvTWu et al. (2021), (vi) PVTWang et al. (2021), (vii) GeminiTeam et al. (2024), and (viii) GPT-4oOpenAI et al. (2024).

The transformer models were implemented using the Huggingface library. The large language models (LLMs) were implemented using few-shot and zero-shot techniques. The zero-shot technique involves providing no examples, while the few-shot technique involves providing a few examples from the dataset. Gemini was implemented using Google's Generative library, whereas GPT was implemented using OpenAI's libraries and API keys.

Hyperparameters used in the models include Learning rate: 1e-4 Batch size: 32 Epochs: 50 Optimizer: AdamW The models were evaluated on a dataset split into 75% training and 25% testing sets.

## 4 EXPERIMENTAL RESULTS

We analyze the performance of various models for detecting mental health-disturbing images, with a primary focus on accuracy as the key metric. The results of our experiments, as presented in Table 2, show that transformer-based models and the Gemini model with different learning strategies (fine-

Table 2: Test results: Detection of Mental health disturbing images. ft(Finetuning), fs(Few Shot) and z(Zero Shot)

| Model | p | r | A |
|---|---|---|---|
| ViT | 62.68 | 61.43 | 62.51 |
| DeiT | 63.90 | 64.45 | 64.23 |
| Swin Transformer | 65.93 | 64.82 | 64.72 |
| TNT | 62.46 | 63.96 | 62.83 |
| CvT | 62.61 | 63.46 | 62.57 |
| PVT | 62.40 | 61.32 | 62.19 |
| Gemini(z) | 57.82 | 58.83 | 59.17 |
| Gemini(fs) | 66.82 | 67.46 | 66.95 |
| GPT-4o(z) | 63.59 | 64.41 | 64.32 |
| GPT-4o(fs) | 68.81 | 69.26 | 70.15 |

tuning, few-shot learning, and zero-shot learning) exhibit varied performance levels. Among the transformer-based models, the DeiT model leads with an accuracy of 64.23%, closely followed by the Swin Transformer with 64.72% accuracy. The Swin Transformer also demonstrates strong performance in terms of precision (65.93%), indicating its robustness across different evaluation metrics. The ViT model, while having the lowest precision at 62.68%, still maintains a competitive accuracy of 62.51%. On the other hand, the PVT model shows the lowest accuracy among the transformer models, with a score of 62.19%, suggesting that it may be less effective for this specific task.

The Gemini model, evaluated in both zero-shot and few-shot settings, reveals a stark contrast in performance. The zero-shot setting yields the lowest accuracy at 59.17%, indicating limited effectiveness without task-specific adaptation. However, when few-shot learning is applied, the Gemini model's accuracy significantly improves to 66.95%, outperforming all transformer-based models except the GPT-4o in the few-shot setting. The GPT-4o model is similarly evaluated in zero-shot and few-shot scenarios. In the zero-shot setting, GPT-4o achieves an accuracy of 64.32%, which is competitive with the highest-performing transformer models. However, in the few-shot setting, GPT-4o excels with the highest accuracy of 70.15%, substantially surpassing all other models. This highlights GPT-4o's exceptional ability to leverage few-shot learning to enhance performance significantly.

Overall, the experimental results demonstrate that while traditional transformer-based models such as DeiT and Swin Transformer provide solid performance, the Gemini and GPT-4o models show remarkable improvements with few-shot learning. Particularly, the GPT-4o model stands out with its superior accuracy, emphasizing the potential of few-shot learning in effectively detecting mental health-disturbing images.

## 5 DISCUSSION

### 5.1 ERROR ANALYSIS

The performance of machine learning models trained on our novel dataset, curated to include distressing content such as gore and graphic materials, was evaluated with an accuracy of 70.15%. We conducted a detailed error analysis focusing on false positives and false negatives.

**False Positives:** False positives occur when the model incorrectly flags non-disturbing images as harmful. Several categories of images contributed significantly to false positives. Medical images, for instance, including surgical procedures, wound treatments, or anatomical illustrations, were often misclassified. These images, although graphic, are typically not intended to disturb the viewer but may visually resemble harmful content, leading to misinterpretation by the model. Similarly, scenes from horror movies designed to appear disturbing but fictional also led to false positives. The model sometimes flagged these images due to their graphic nature, mistaking them for real distressing content. Additionally, images of artistic and theatrical makeup, where individuals wear elaborate

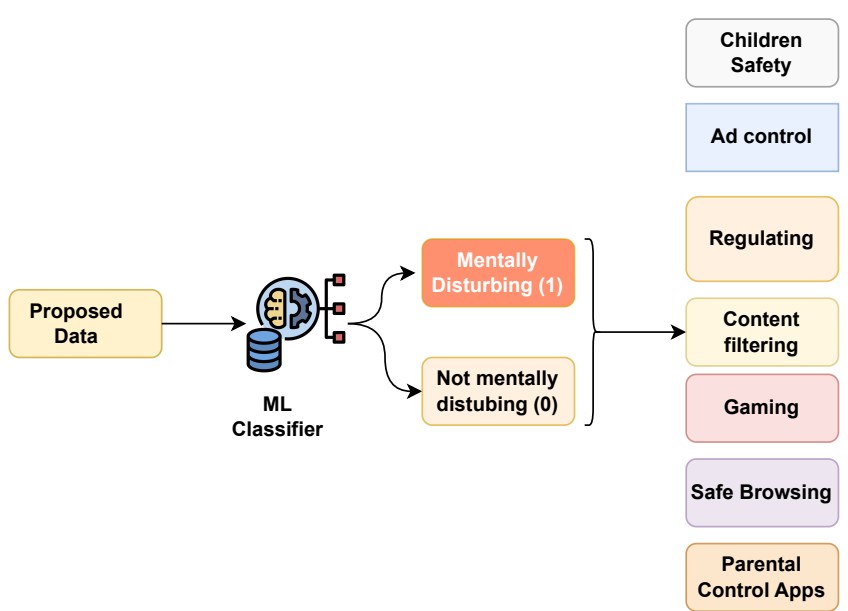

Figure 2: Real time applications of the proposed classifier

makeup mimicking injuries or deformities for artistic purposes, were frequently misclassified. The realistic appearance of these makeup effects confused the model, causing it to label them as harmful.

**False Negatives:** False negatives, where the model fails to identify genuinely harmful images, present another significant challenge. Certain types of images were commonly misclassified in this manner. Images with subtle distress, such as those showing neglected animals or minor injuries, often went unflagged. The less overt nature of these images made it difficult for the model to recognize them as harmful, as they lacked the explicit graphic elements it was trained to detect. Composite images, where harmful content is embedded within a larger, non-disturbing context, also contributed to false negatives. The presence of benign elements diluted the impact of the harmful content, causing the model to overlook the disturbing parts. Additionally, low-quality or blurred images of distressing scenes were frequently misclassified. The poor resolution and lack of clarity in these images hindered the model's ability to accurately detect harmful elements.

## 5.2 Practical Applications and Usecases

The proposed classifier, which detects and filters mentally disturbing images on social media, has several real-time applications and deployment possibilities across various platforms and industries as shown in Figure 2. Here are some key use cases

1. **Children Safety:** Protect students using online learning platforms from encountering harmful images, especially in forums and discussion boards.

2. **Ad Content Control:** Ensure advertisements do not include graphic content, maintaining brand safety and audience appropriateness.

3. **Government Regulations:** The government bodies can use the proposed classifier to identify mentally disturbing content and regulate them. Some of the social media platforms do not consider regulating.

4. **Content Filtering:** Automatically flag and blur or remove graphic images from news articles to make content more appropriate for general audiences.

5. **Safe Browsing:** Implementing the algorithm in browsers to detect and warn users about graphic content on web pages.

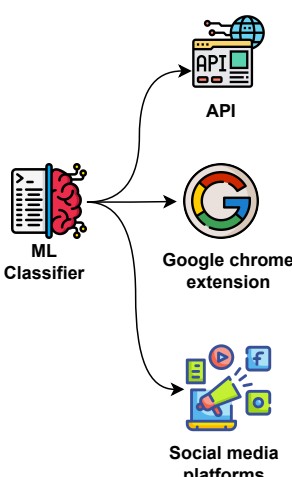

Figure 3: Real time deployments of the proposed classifier

6. **Parental Control Apps:** Help parents control the type of content their children can access, preventing exposure to graphic images.

7. **Gaming:** Monitor and regulate user-generated content in gaming communities, such as in-game screenshots or shared images, to prevent the spread of harmful images.

## 5.3 REAL TIME DEPLOYMENTS

There are several deployments of the proposed classifier for supporting real time applications as presented in figure 3. They are:

**Chrome Extension for Social Media Filtering:** Deploying the algorithm as a Chrome extension would allow users to filter out distressing images while browsing social media platforms. The extension could analyze images in real-time as users scroll through their feeds, blurring or removing harmful content before it is displayed. This would provide a safer browsing experience, reducing the risk of exposure to graphic content.

**API for Social Media Platforms:** Social media companies could integrate the algorithm into their platforms through an API. This deployment would allow these platforms to automatically scan and filter user-uploaded images for graphic content, enhancing content moderation processes and protecting users from harmful visuals. The API could be used during the image upload process or continuously monitor posted content.

**Deploying in Social media platforms:** Deploying the classifier in the social media platforms for regulating the creators to post mentally disturbing content.

## 6 LIMITATIONS

Despite the promising results and potential impact of this work, several limitations must be acknowledged. Firstly, the dataset used in this study is entirely sourced from Reddit. While Reddit is a significant and diverse platform, its content and user base may not be fully representative of other social media platforms. This reliance on a single source may limit the generalizability of the models developed using this dataset. Therefore, future research should consider incorporating data from multiple social media platforms to enhance the robustness and applicability of the findings.

Another important limitation is the absence of textual context. The dataset and subsequent models focus solely on visual content, specifically graphic and distressing images. In real-world scenarios, such images are often accompanied by textual content that can provide crucial context for understanding the severity and nature of the content. The lack of textual context in this study means that the models may not perform as effectively when deployed in environments where text and images are intertwined. Incorporating multimodal data, including both images and text, in future research could improve the accuracy and utility of content filtering models.

## 7 ETHICAL CONSIDERATIONS

The creation and use of a dataset containing distressing and harmful images necessitate careful ethical considerations to ensure the responsible handling and application of this sensitive content. First and foremost, we firmly stand against any misuse of the dataset. The dataset is intended solely for the development of machine learning models aimed at identifying and filtering harmful content to protect users' mental well-being on social media platforms. Unauthorized or malicious use of this dataset, such as for the propagation of harmful content, exploitation, or any activities that could cause distress or harm to individuals, is strictly prohibited.

Moreover, we have taken several measures to mitigate potential ethical risks. The collection and curation of images were conducted with utmost respect for privacy and copyright laws, ensuring that no personal or identifiable information is included. Additionally, access to the dataset is restricted to researchers and developers committed to creating safer online environments, with clear guidelines and usage agreements in place to prevent misuse.

We also emphasize the importance of transparency and accountability in the research and development process. All researchers and developers utilizing this dataset are encouraged to adhere to ethical guidelines, seek appropriate institutional approvals, and engage in continuous dialogue about the ethical implications of their work. By promoting responsible use and safeguarding against misuse, we aim to contribute positively to the field of content moderation and the broader goal of enhancing user safety and mental health in digital spaces.

## 8 CONCLUSION AND FUTURE WORK

We introduce a novel dataset specifically curated to include mentally disturbing images from social media, with the aim of facilitating the development of machine learning models capable of detecting and filtering such harmful content. By leveraging this dataset, we developed classifiers that achieved an accuracy score of 70.15%, demonstrating their potential for deployment in real-time scenarios to identify and flag distressing images.

The further contributions which can improve the work could be expanding the dataset to include content from multiple social media platforms is essential. Platforms like Twitter, Facebook, Instagram, and TikTok have diverse user bases and content types that could enrich the dataset and improve the generalizability of the developed models. By incorporating data from various sources, future models could be better equipped to handle a wider array of graphic content and improve overall detection accuracy. Integrating multimodal data, including both visual and textual content, represents a crucial step forward. Images on social media often come with captions, comments, and other textual elements that provide important context. By training models on both image and text data, future research can develop more sophisticated algorithms capable of understanding and filtering harmful content more effectively. This approach would likely reduce false positives and enhance the models' ability to discern the context in which an image is presented.

We release our code and dataset at `https://xxxxxxxxxxxx`

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

## APPENDIX

## A HOW ARE NSFW DATASETS DIFFERENT FROM THE PROPOSED DATASET ?

### 1. CONTENT TYPE

- **NSFW Datasets**: These primarily include explicit adult content such as nudity, pornography, or sexually suggestive material. The focus is often on sexual themes, and the primary concern is preventing exposure in inappropriate settings.

- **Mentally Disturbing Image Datasets**: These include images that could cause psychological discomfort, such as graphic violence, horror, abuse, self-harm, or trauma-inducing scenes. The content may evoke strong negative emotions like fear, disgust, or distress.

## 2. PURPOSE AND USE

- **NSFW Datasets**: These datasets are often used in content moderation, filtering algorithms, or for research in detecting adult content. The goal is to identify and restrict inappropriate sexual content in platforms where it does not belong, such as workplaces, social media, or public forums.

- **Mentally Disturbing Image Datasets**: These datasets are typically used in fields like forensic analysis, trauma research, or for developing AI systems that can detect harmful or violent content. They are used to prevent the spread of such images or for law enforcement purposes, but handling them requires more ethical sensitivity due to the potential psychological impact.

## 3. ETHICAL AND PSYCHOLOGICAL CONSIDERATIONS

- **NSFW Datasets**: The primary ethical concerns involve respecting privacy, ensuring consent, and preventing access to adult content by unintended audiences, such as minors. Psychological effects are generally less severe, though non-consensual imagery can lead to more significant concerns.

- **Mentally Disturbing Image Datasets**: These datasets raise more serious ethical issues due to their potential to cause trauma or distress to viewers. The content could have lasting psychological impacts, particularly if viewed without adequate preparation or mental health support.

## 4. LEGAL IMPLICATIONS

- **NSFW Datasets**: Legal concerns primarily involve preventing access to explicit adult material by minors or in public environments. The specific regulations vary by country, and content moderation systems need to comply with these legal frameworks.

- **Mentally Disturbing Image Datasets**: These datasets may contain illegal content (e.g., graphic violence, abuse) and therefore require even stricter legal safeguards. Access is often restricted to authorized personnel, such as law enforcement, due to the severity of the content.

## 5. AUDIENCE SENSITIVITY

- **NSFW Datasets**: These datasets aim to protect the general public from unexpected exposure to adult content. Although controversial, the psychological impact of such content may not be as severe as that of violent or disturbing imagery.

- **Mentally Disturbing Image Datasets**: These images can have severe psychological effects, particularly on individuals with pre-existing mental health conditions or trauma. Special precautions must be taken to limit access and ensure viewers are properly informed of potential harm.

