# OpenReview forum: "From Eww to Woo: Detection of Mental Health Disturbing Images in Social Media"
_ICLR.cc/2025/Conference — ICLR 2025 Conference Withdrawn Submission_

### Official Review · Reviewer_cj4o · 2024-10-27

**Soundness:** 2
**Presentation:** 2
**Contribution:** 3
**Rating:** 3
**Confidence:** 3

**Summary:**

The paper, titled "From Eww to Woo: Detection of Mental Health Disturbing Images in Social Media," introduces a new dataset aimed at identifying and filtering mentally distressing images on social media platforms. This dataset includes graphic content, such as gore, which is commonly linked to negative mental health impacts like anxiety, depression, and PTSD. The authors evaluated this dataset using various advanced machine learning models, achieving an accuracy of 70.15%. The importance of this work lies in its potential to enhance online safety by effectively detecting harmful visual content, thereby reducing mental health risks for users.

**Strengths:**

1. This paper offers practical implications, especially for enhancing mental health safety on social media platforms.
2. Experts with psychology backgrounds annotated the dataset, ensuring the labels accurately capture the images' potential psychological effects. This expert annotation adds credibility and reliability, establishing the dataset as a valuable resource for future studies.
3. Developing a specialized dataset primarily targeting mentally disturbing images represents a major contribution.

**Weaknesses:**

1. Although the dataset is innovative, it might lack the diversity and scope needed for broad generalization across different types of disturbing content or varying cultural perspectives on what constitutes mentally distressing imagery. Expanding the dataset to cover a wider range of content and contexts could enhance its usefulness.
2. The models applied to the dataset achieved an accuracy of 70.15%, which, while commendable, may fall short of practical application needs. If the authors believe this accuracy is adequate, could they clarify the reasoning behind this?

**Questions:**

1. The dataset is mainly sourced from Reddit and Google. It indicates that the dataset’s representativeness be affected by relying heavily on these sources. Could this limit its generalizability to other platforms?
2. Were cultural variations considered in the annotation process, as different cultures may have varying thresholds for what constitutes mentally disturbing?
3. How consistently can annotators label content as mentally disturbing or non-disturbing, given the subjective nature of emotional impact?
4. Images are often accompanied by text that provides essential context. How might the exclusion of such contextual information impact the model's effectiveness?
5. The best-performing model achieves an accuracy of 70.15%. For deployment in a real-world setting, what measures could be taken to reduce potential errors given this accuracy level?
6. The results show a significant accuracy increase with few-shot learning. What are the potential limitations or scalability issues in deploying few-shot models in real-time content moderation?
7. Were psychological support or debriefing sessions provided to annotators, given the emotionally challenging nature of the content? If not, what are the potential long-term effects on annotators, and should future projects include mental health resources?
8. Could this model be adapted for other use cases, such as trauma recovery or resilience training? What potential ethical risks might arise from such applications, particularly in terms of user consent and sensitivity?

---

### Official Review · Reviewer_Fp3W · 2024-11-02

**Soundness:** 2
**Presentation:** 2
**Contribution:** 2
**Rating:** 5
**Confidence:** 3

**Summary:**

This paper addresses an urgent concern in digital well-being by introducing a novel dataset and machine learning models aimed at detecting and filtering harmful images on social media. The dataset, curated from platforms like Reddit, includes images that could be psychologically disturbing, such as gore and graphic content, aiming to support machine learning models designed for content moderation. Using state-of-the-art models such as transformers and language models (LLMs), the study benchmarks the effectiveness of these algorithms, with the GPT-4o model achieving the highest accuracy at 70.15% under few-shot settings. The paper emphasizes potential real-world applications in safeguarding mental health by integrating the model into social media platforms, ads, and parental control applications.

**Strengths:**

- The paper tackles a pressing issue in social media moderation, which contributes to user safety and mental well-being.
- By curating a dataset focused on disturbing content, this research fills a gap in the resources available for content moderation research. The balanced distribution of labels in training and test sets strengthens the robustness of the dataset.
- The comparative analysis of multiple models, particularly in both few-shot and zero-shot settings, provides insights into the potential and limitations of current machine learning approaches in detecting disturbing visual content.
- The paper includes an error analysis, identifying common sources of false positives (such as medical images and horror scenes) and false negatives (subtle distress or low-quality images), which is valuable for refining future model development.

**Weaknesses:**

- The study primarily benchmarks transformer-based models and a few language models. While this covers popular architectures, it could be strengthened by testing a broader range of model types, including convolutional neural networks (CNNs), which have proven effective in image classification, particularly in distinguishing nuanced visual features.
Reliance solely on Reddit limits the generalizability of the findings. Expanding data sources to other platforms would enhance the dataset's representativeness.
- The model's focus on visual content may reduce its real-world applicability, as text often accompanies such images and provides necessary context for interpretation. Incorporating multimodal data in future research could improve detection accuracy.
- While noteworthy, the best-performing model achieved an accuracy of 70.15%, which still highlights room for improvement, especially in a high-stakes field like content moderation.
- While the models are evaluated on the curated dataset, there’s no testing in real-world social media environments. Real-world data often contains noise, varied image resolutions, and mixed-content types, which could impact model performance. Without testing in a realistic setting, it's hard to assess how well these models would perform on social media at scale.
- While the few-shot and zero-shot approaches for GPT-4o and other models are mentioned, the paper lacks a detailed analysis of why few-shot learning significantly improves performance for some models and how this could be leveraged in future applications. This could be important for deployment where labelled data may be limited.
- Although the paper provides a general overview of annotation guidelines, the definitions of “mentally disturbing” vs. “non-disturbing” images are somewhat subjective and open to interpretation. The criteria for labelling could benefit from additional clarity or examples, as this may impact the consistency of annotations and, by extension, model accuracy.

**Questions:**

- By sourcing data from specific Reddit channels and Google searches, the dataset may contain inherent biases related to the types of content commonly shared on these platforms. This could affect model generalization, especially if certain distressing images are overrepresented or underrepresented.
- Content disturbing in one culture may not be perceived similarly in another. The dataset and models do not account for cultural differences in what is considered disturbing, which may limit the tool’s applicability across diverse user bases on global platforms.
- Model interpretability is crucial for content moderation tools, especially those affecting user experience. The paper does not address how or if the models explain their decisions, which could be important for transparency, especially if incorrect flagging affects user content visibility.

**Details Of Ethics Concerns:**

- While ethical considerations are mentioned, the paper does not specify safeguards against misuse by unintended actors who may access the dataset, such as content creators looking to bypass content moderation. A more robust dataset-sharing framework, including a licensing system and access control, could help mitigate potential misuse.
- Although ethical guidelines exist, exposure to distressing content for annotators still poses psychological risks. The authors do not discuss any specific support mechanisms or periodic assessments of annotators’ well-being, which would be crucial in a project involving prolonged exposure to disturbing content.

---

### Official Review · Reviewer_PHgm · 2024-11-03

**Soundness:** 2
**Presentation:** 2
**Contribution:** 2
**Rating:** 3
**Confidence:** 4

**Summary:**

The authors present a new dataset sourced from Reddit and Google search results that contains images that are disturbing for mental health. Additionally, they test a number of state-of-the-art computer vision models and demonstrate the challenging nature of the task at hand.

**Strengths:**

- The paper addresses an important problem with significant societal impact. Developing an accurate detection model for such kind of images could be valuable and useful in a number of applications.
- The authors present an insightful discussion of the different types of errors in their empirical analysis.

**Weaknesses:**

- The related work discussion includes a number of works that are only tangentially related to the topic of the paper. For instance, detecting tampered and fake images is rather irrelevant to the topic of the paper.

- At the same time, the paper misses some highly relevant works [1, 2] that could enrich the related work discussion and potentially the empirical analysis.

- The error analysis reveals that there is a lot of room for personal interpretation regarding what is considered disturbing. It would be worth reporting more meticulously on the prevalence on the different types of error.

- The experimental section is rather limited in terms of studied options/setups/ablations/etc. The resulting accuracy levels are not very impressive (which is somewhat understandable due to the challenging and subjective nature of the task).

- The ethical issues raised by the paper are not properly addressed (cf. below).

- The writing quality could be improved both in terms of clarity and in terms of style and elimination of errors.


[1] Sarridis, et al. (2022, December). Leveraging large-scale multimedia datasets to refine content moderation models. In 2022 IEEE Eighth International Conference on Multimedia Big Data (BigMM) (pp. 125-132). IEEE.

[2]  Zampoglou, et al. (2016, December). A web-based service for disturbing image detection. In International Conference on Multimedia Modeling (pp. 438-441). Cham: Springer International Publishing.

**Questions:**

I have two main suggestions:
- revisit related work in the area and carefully consider the implications for this work
- reconsider the ethical issues

**Details Of Ethics Concerns:**

While the authors acknowledge the numerous ethical issues in their work, they seem to address them only in a superficial manner. My primary concern is the emotional well-being of the three annotators who undertook the task of examining and annotating 17,776 images. This seems highly risky and intensive and simply instructing the annotators to take breaks does not seem sufficient as a precautionary measure.

Other than that, it is not clear how the authors respect the copyright of the dataset images given that they are sourced from Reddit and the web, and it's unlikely that the authors requested permission for their use. A similar concern arises regarding the privacy issues that might arise.

---

### Official Review · Reviewer_SCnK · 2024-11-04

**Soundness:** 2
**Presentation:** 2
**Contribution:** 2
**Rating:** 5
**Confidence:** 4

**Summary:**

The authors present a novel dataset specifically curated to include mentally disturbing images from social media, with the aim of facilitating the development of machine learning models capable of detecting and filtering such harmful content. They developed classifiers that achieved an accuracy score of 70.15%, demonstrating their potential for deployment in real-time scenarios to identify and flag distressing images.

The paper lacks in following areas:

1. Model Performance: The reported accuracy (70.15%) indicates room for improvement, as this level may not be sufficient for practical deployment. Further experimentation with more sophisticated architectures or ensemble techniques could yield higher accuracy and robustness.

2. Ethical and Privacy Considerations: Although the study briefly touches on the mental health risks of exposure, more discussion is needed regarding the ethical implications of collecting and sharing graphic content, particularly in terms of privacy and potential retraumatization during model training and evaluation.

3. Comparative Analysis: While the authors provide benchmarks on state-of-the-art models, the paper could benefit from a comparative analysis with similar datasets or models designed for related tasks (e.g., detecting NSFW or violent content). This comparison would help contextualize the dataset’s performance and further highlight its unique contributions.

4. Broader Model Applicability: The study would be strengthened by discussing potential applications beyond social media platforms, such as in law enforcement or content moderation for media platforms. This addition would broaden the appeal of the work and underline its relevance across various domains.

The paper has a potential for improvement and needs more work.

**Strengths:**

The paper presents:
1. Novel Dataset: The dataset introduced is a notable advancement, specifically curated to include a broad range of graphic and mentally disturbing images. Unlike general-purpose datasets, this dataset provides a targeted resource for developing and benchmarking models focused on detecting harmful content.

2. Annotation Quality: The annotation process is rigorous, with images reviewed by annotators trained in psychology, lending reliability and psychological validity to the classification of images as either mentally disturbing or non-disturbing. This approach helps ensure the dataset's relevance and quality for real-world applications.

3. Bench marking and Model Performance: The paper benchmarks the dataset on state-of-the-art models, achieving an accuracy of 70.15%, which, while not exceptionally high, indicates a functional baseline for future improvements. This initial performance suggests that the dataset can support the development of models with reasonable efficacy in identifying distressing content.

4. Relevance and Potential Impact: The study addresses a critical gap in content moderation technology. By improving the detection of disturbing images, the proposed work has the potential to help social media platforms create safer environments, thereby reducing mental health risks for users exposed to graphic content.

**Weaknesses:**

The paper lacks in following areas:

1. Model Performance: The reported accuracy (70.15%) indicates room for improvement, as this level may not be sufficient for practical deployment. Further experimentation with more sophisticated architectures or ensemble techniques could yield higher accuracy and robustness.

2. Ethical and Privacy Considerations: Although the study briefly touches on the mental health risks of exposure, more discussion is needed regarding the ethical implications of collecting and sharing graphic content, particularly in terms of privacy and potential retraumatization during model training and evaluation.

3. Comparative Analysis: While the authors provide benchmarks on state-of-the-art models, the paper could benefit from a comparative analysis with similar datasets or models designed for related tasks (e.g., detecting NSFW or violent content). This comparison would help contextualize the dataset’s performance and further highlight its unique contributions.

4. Broader Model Applicability: The study would be strengthened by discussing potential applications beyond social media platforms, such as in law enforcement or content moderation for media platforms. This addition would broaden the appeal of the work and underline its relevance across various domains.

**Questions:**

The paper has a potential for improvement and needs more work.

---

### Note · Authors · 2024-11-23

**Comment:**

We decided to submit to other conference

**Withdrawal Confirmation:**

I have read and agree with the venue's withdrawal policy on behalf of myself and my co-authors.